# Exploring the Role of Circulating Cell-Free RNA in the Development of Colorectal Cancer

**DOI:** 10.3390/ijms241311026

**Published:** 2023-07-03

**Authors:** Chau-Ming Kan, Xiao Meng Pei, Martin Ho Yin Yeung, Nana Jin, Simon Siu Man Ng, Hin Fung Tsang, William Chi Shing Cho, Aldrin Kay-Yuen Yim, Allen Chi-Shing Yu, Sze Chuen Cesar Wong

**Affiliations:** 1Department of Health Technology and Informatics, The Hong Kong Polytechnic University, Hong Kong SAR, China; kantrevor@gmail.com (C.-M.K.); andy_thf@yahoo.com.hk (H.F.T.); 2Department of Applied Biology & Chemical Technology, The Hong Kong Polytechnic University, Hong Kong SAR, China; xiaomeng.pei@connect.polyu.hk (X.M.P.); martin-ho-yin.yeung@polyu.edu.hk (M.H.Y.Y.); 3Codex Genetics Limited, Shatin, Hong Kong SAR, China; jinnana7926@hotmail.com (N.J.); aldrinyim@codexgenetics.com (A.K.-Y.Y.); allenyu@codexgenetics.com (A.C.-S.Y.); 4Department of Surgery, Faculty of Medicine, The Chinese University of Hong Kong, Hong Kong SAR, China; simon.ng@cuhk.edu.hk; 5Department of Clinical Oncology, Queen Elizabeth Hospital, Kowloon, Hong Kong SAR, China; williamcscho@gmail.com

**Keywords:** circulating tumor RNA, circulating cell-free RNA, colorectal cancer, biomarker discovery, blood-based liquid biopsy, circulating RNA sequencing, diagnostic, prognostic and predictive biomarkers of CRC

## Abstract

Circulating tumor RNA (ctRNA) has recently emerged as a novel and attractive liquid biomarker. CtRNA is capable of providing important information about the expression of a variety of target genes noninvasively, without the need for biopsies, through the use of circulating RNA sequencing. The overexpression of cancer-specific transcripts increases the tumor-derived RNA signal, which overcomes limitations due to low quantities of circulating tumor DNA (ctDNA). The purpose of this work is to present an up-to-date review of current knowledge regarding ctRNAs and their status as biomarkers to address the diagnosis, prognosis, prediction, and drug resistance of colorectal cancer. The final section of the article discusses the practical aspects involved in analyzing plasma ctRNA, including storage and isolation, detection technologies, and their limitations in clinical applications.

## 1. Introduction 

Colorectal cancer (CRC) has the second-highest rate of new diagnoses and cancer-related deaths worldwide, with nearly 1,930,000 and 930,600 cases in 2020, respectively [1]. According to the worldwide surveillance of cancer survival within the CONCORD program, the 5-year survival rate for people with CRC is approximately 71% in Southeast Asia, with large variations depending on the stage of the disease at the time of diagnosis—from approximately 90% for those diagnosed at stage I to slightly over 10% for those who develop metastatic cancer at stage IV [2,3,4]. The surgical resection of CRC is the standard of care, followed by adjuvant chemotherapy based on the clinical and pathological risk factors for the disease [4]. Despite this, relapse still occurs in more than 30% of patients with resectable CRC [4]. The current approaches, such as periodic computed tomography scans and carcinoembryonic antigen (CEA) level monitoring, demonstrate limited sensitivity to detect recurrent disease [5]. Given this, there is an urgent need to develop methods that are cost-effective, sensitive, and accurate for the detection of minimal residual disease (MRD), which can inform treatment decisions, particularly for patients with a borderline performance status who are concerned about treatment side effects [4,6].

CRC is a complex disease that can be influenced by a variety of genetic and epigenetic factors. It has been demonstrated that the widely accepted genetic model of tumor development, which involves mutations in specific genes such as adenomatous polyposis coil (APC), Kirsten-ras, (K-ras), and p53, is unrepresentative of the majority of CRC cases [7]. Multiple genetic pathways have been found to contribute to CRC, resulting in a heterogeneous pattern of tumor mutations [7]. The development of colorectal cancer has also been linked to epigenetic changes, including abnormal DNA methylation and chromatin modification [8]. CRC types with methylated genes have been classified as CpG Island Methylator Phenotype (CIMP) cancers, which are characterized by a high frequency of methylated genes [8]. High intra-tumor heterogeneity (ITH) and inter-patient heterogeneity are well-known features of CRC that impact tumor molecular characterization [9]. These mutations are composed of multiple cell types or subclones, each of which shows different gene expression profiles [9]. Additionally, some studies have found that exogenous factors such as diet, lifestyle, nutrition, the microbiome, and the environment can affect pathogenesis, as well as non-neoplastic cells, such as immune cells, resulting in further heterogeneity [9,10]. Due to all these differences, different patients respond differently to treatment and have varying outcomes. Therefore, a growing number of researchers have now recognized the importance of investigating the interactions between tumor molecular changes and the tumor microenvironment (TME), which vary from individual to individual [11]. With the development of high-sensitivity detection and quantification techniques, such as next-generation sequencing (NGS), genome-wide sequencing, and droplet digital polymerase chain reactions (ddPCR), the study of heterogeneity has become more feasible. Currently, whole-exome sequences performed on multiple biopsy sites are the standard method of assessing ITH. These analyses are subject to the limitations of the tissue biopsy, such as being non-repeatable, providing limited information from a single biopsy site, low frequency, and inaccuracy, leading to the underestimation of the mutational landscape, which may affect the treatment accuracy [12,13].

To overcome the limitations of tissue biopsies, liquid biopsies, as a minimally invasive approach to the analysis of genetic material in the blood, serum, plasma, urine, and saliva [14,15,16,17,18,19], have rapidly developed in recent years. Liquid biopsies involve isolating tumor-derived components, such as circulating tumor cells (CTCs), circulating tumor DNA (ctDNA) and RNA (ctRNA), and extracellular vesicles (EV), which can provide additional useful information for diagnostic, prognostic, predictive, drug resistance [20,21,22], and subtype classification [23] purposes according to their multi-omics data, including genomics, transcriptomics [24], proteomics [13], and metabolomics [25]. For instance, a liquid biopsy can be used to detect the recurrence of CRC and monitor the treatment response [26], as well as to detect mutations, such as *RAS* mutations, which are associated with treatment resistance and disease progression [27].

The circulating RNA sequencing technique has been widely used in clinical research to identify circulating cell-free RNA (cfRNA) biomarkers that may be associated with a variety of diseases, including infectious diseases, cancer, and autoimmune disorders. Researchers can gain insights into the molecular mechanisms behind disease development and progression by analyzing the RNA profiles of cfRNA. cfRNA is released from cells into the extracellular matrix and primarily serves as a signaling molecule in cell-to-cell communication [14]. In a similar manner to ctDNA, cfRNA is passively leaked from apoptotic, tumor, or necrotic cells [28]. cfRNA was first discovered in 1999 in the plasma of nasopharyngeal carcinoma patients [29], followed by its discovery in the serum of melanoma patients [30]. In 2006, Wong et al. were the first to demonstrate that plasma beta-catenin mRNA can serve as a potential marker for CRC [31]. Recently, cfRNAs have been recognized as biomarkers that are useful in identifying and detecting tumors and monitoring personalized therapies [28,32].

In this work, we present an up-to-date overview of the current knowledge of cfRNAs and their status as biomarkers to address the diagnosis, prognosis, prediction, and drug resistance of CRC. Finally, we conclude by discussing the practical aspects of the analysis of cfRNA, including storage and isolation, detection technologies, and the limitations of clinical applications.

## 2. Blood-Based Liquid Biopsy in CRC

In recent years, increased interest has emerged in various circulating biomarkers, such as CTCs, and various forms of EV/platelet-encapsulated, ctDNA, and ctRNA, including messenger RNA (mRNA), microRNA (miRNA), circular RNA (circRNA), and long non-coding RNA (lncRNA) [33]. The strengths and limitations of different liquid biopsy components are listed in Table 1 [32,34,35].

CTCs are tumor cells that have detached from the primary tumor and entered the bloodstream, and they can be isolated from peripheral blood and analyzed for DNA, RNA, and protein markers [36]. Our previous study effectively demonstrated that the count of CTCs varies in relation to the tumor node metastasis (TNM) stage, as well as between pre-operative and post-operative phases. This result suggests their utility in tracking therapy responses [37] and offering prognostic and potentially predictive value [38,39,40] for patients with metastatic CRC. CTCs have been demonstrated in several studies to be associated with a poor prognosis and short overall survival in patients with CRC [41] and in the monitoring of metastatic process and disease progression [38,39,40,42]. Furthermore, CTCs have the potential to provide valuable insights into the biology of CRC and the mechanisms of metastasis. A detailed analysis of the genomic and phenotypic characteristics of CTCs can provide information regarding potential targets for therapy, as well as a better understanding of the mechanisms underlying tumor progression and metastasis [42]. In spite of these strengths, CTCs still have several limitations in CRC research and clinical practice. The rarity of CTCs in peripheral blood, being present at very low concentrations, makes their detection and isolation difficult [43]. Currently, CTC detection methods are limited in their sensitivity, and some patients may lack the presence of CTCs [44]. Moreover, the heterogeneity of CTCs poses a challenge to their characterization and analysis. The clinically relevant and useful functions of CTCs may be influenced by their phenotypic and molecular heterogeneity [45]. Besides this, the detection and analysis of CTCs are also limited by the absence of standard methods [46]. Due to methodological constraints, CTCs have not yet gained widespread acceptance as a crucial component of cancer patient care. Such limitations may result in the requirement for substantial quantities of fresh blood and labor-intensive, expensive processes [47]. This obstacle is especially prominent in the early stages of cancer, considering the minimal number of CTCs present in the cancer patient’s bloodstream [48,49]. In addition, false positives may occur when CTCs are detected in inflammatory diseases of the gastrointestinal tract. For example, the chronic inflammation of the gastrointestinal tract, as a characteristic of inflammatory bowel disease (IBD), can lead to the massive infiltration of circulating leukocytes into the intestinal tract, resulting in ulcerative colitis and Crohn’s disease. These leukocytes, such as neutrophils, T-cells, and monocytes, can cause inflammation and contribute to false positive results [50]. False negative results can occasionally occur when circulating tumor cells (CTCs) are isolated and detected [51,52].

EVs have emerged as important players in CRC research, with both strengths and limitations. EVs, including exosomes, contain various molecules that can serve as biomarkers to assess the diagnosis, prognosis, and response to treatment for CRC [53]. In addition to reflecting tumor heterogeneity, they also play a role in intercellular communication, which affects tumor progression, metastasis, and the immune response [53]. The isolation of EVs from various body fluids provides an effective method for non-invasive sampling. In spite of this, the heterogeneity of EVs and the lack of standardization of protocols for their isolation and characterization pose considerable challenges. Other limitations include contamination with non-EV particles and difficulties in determining the functions of the particles. The regulation of enriched microRNA by tumor-derived EVs has been found to promote tumor progression and T-cell dysfunction [54]. A high level of circulating EVs has been associated with a poor prognosis in CRC patients and a shorter survival time, as they stimulate and suppress tumor-specific and non-specific immune responses [54]. Based on these findings, EVs may contribute to tumor progression by modulating the immune response. However, the heterogeneity of EVs and the lack of standardized protocols for their isolation and characterization pose challenges [53]. Contamination with non-EV particles and functional characterization difficulties are additional limitations [53]. 

The concentration of circulating cell-free nucleic acid is associated with the aggressiveness of CRC, disease stage, and tumor volume, indicating the treatment response, and can also serve as a prognostic marker in evaluating whether the disease is progressive, relapsing, or MRD [20,21,22,55]. With the usage of high-sensitivity detection and quantification techniques, in clinical research and practice, circulating cell-free DNA (cfDNA) has been increasingly applied in the detection of resistance mutations and oncogenic driver mutations [20,21,22,55]. For instance, mutations in *APC*, *KRAS*, *TP53*, and *SMAD4* have been reported as key drivers of progression and metastasis in CRC [56]. Besides this, the number of resistance mechanisms to anti-EGFR therapies in CRC patients has been previously reported, including mutations of *BRAF*, *MEK,* and the *EGFR* extracellular domain (ECD) and the amplification of *ERBB2*, *MET*, *KRAS,* and *NRAS*, which could benefit from the inclusion of targeted therapies in standard protocols, emphasizing the importance of personalized medicine [20,21,22]. Recently, researchers identified acquired genomic alterations, including mutations in *RAS*, *BRAF*, and the *EGFR* ectodomain, as the major mechanisms of resistance to later-line anti-EGFR antibody therapy in metastatic CRC patients who received cetuximab or bevacizumab as major mechanisms of resistance to early-stage therapies [57].

Although the tumor burden is positively correlated with ctDNA, false negative results arising from small ctDNA concentrations, the short half-lives of ctDNA, and low signal-to-noise ratios make it difficult to detect cfDNA and determine the tumor tissue of origin (TOO) [24], especially in the early stages, when metastasis has begun [12,58,59]. Another practical challenge regarding cfDNA is that the majority of cfDNA features are not tissue-specific, which makes it difficult to determine the tumor tissue of origin of a patient who has been positively screened for cancer [14]. ctDNA analysis is also limited by the lack of standardization of methodologies [59]. Furthermore, ctDNA analysis is currently unable to analyze RNA [59]. Although the targeted analysis of methylation markers on cfDNA can cause cancer to be localized in a highly specific manner [60], it is important to explore additional biomarkers to complement detection by cfDNA in the early stages of cancer, to enhance its detection and localization. Considering that cfRNA analysis provides valuable information on gene expression, splicing, and post-transcriptional regulation, cfRNA become an additional biomarker for the diagnosis, monitoring, and treatment of cancer, along with cfDNA.

The stability of cfRNA is relatively low as it is rapidly degraded by ribonucleases and 99% of naked RNA is degraded after only fifteen seconds of incubation [12,61]. However, endogenous cfRNA can be secreted from both cancerous and non-cancerous cells via microvesicles, nucleoproteins, and protein–RNA complexes, which act as protectors to prevent the cfRNA from being degraded by ribonucleases [12,28]. Several studies have demonstrated the recovery of full-length mRNA from plasma, suggesting that cfRNA is relatively stable in the blood [14]. These findings have generated considerable interest in cfRNA as a potential diagnostic biomarker for sensitive, fast, and inexpensive diagnostics in CRC [12,55,62]. 

The use of cfRNA analysis in CRC has several strengths and limitations. One of the major advantages of cfRNA analysis is its ability to profile RNA expression, which provides valuable information about gene expression patterns and potential biomarkers in CRC [24]. Secondly, an analysis of cfRNA within the tumor microenvironment can provide valuable insights into cell-to-cell communication in CRC. Certain cfRNAs may indicate intercellular communication between cancer cells, non-cancerous cells, and microorganisms [63]. For example, extracellular vesicle (EV)-associated non-coding RNAs, including cfRNAs, have been implicated in tumorigenesis and have been shown to serve as diagnostic and therapeutic targets for cancer [63]. Additionally, cfRNA analysis can assist in identifying the tissue of origin (TOO) and cancer subtypes through cell-type decomposition and cell-type-specific RNA markers [14,64]. For personalized medicine and treatment decisions, this information is crucial, as different cancer types may require different therapeutic approaches [64]. Scientists have also used cfRNA analysis to examine the immune heterogeneity associated with particular mutations and tumor microenvironment characteristics, in order to identify immune-related pathways and potential therapeutic targets based on the analysis of cfRNA transcriptomic profiles [65,66]. However, there are limitations associated with cfRNA analysis in CRC. Firstly, the amount of cfRNA is low, making it difficult to detect and analyze it [67]. Furthermore, cfRNA is susceptible to degradation and instability, which further complicates its analysis [67]. Obtaining cfRNA from blood samples can be challenging, requiring specialized techniques to isolate and preserve the RNA molecules [68]. Additionally, there is a lack of standardization of procedures in cfRNA analysis, resulting in inconsistent results across different studies and laboratories [68]. Due to these limitations, cfRNA analysis cannot be widely used and implemented in the clinical setting for CRC.

A higher level of cfRNA has been observed in patients with solid tumors [69]. It has been suggested that cancer cells communicate with the surrounding immune and stromal cells via extracellular RNA, which can lead to the increased proliferation and malignancy of surrounding cells, angiogenesis [70], and the development of future metastatic sites due to escaping the immune response [71,72]. As a biomarker, cfRNA may hold certain advantages, despite being technically more challenging than cfDNA. The ability of cfRNA to play a role in mediating intercellular communication may allow us to improve our understanding of the intercellular communication pathways involved in the normal differentiation as well as the initiation and transformation of CRC. The use of cfRNA has been shown to increase the yield of gene expression information, especially when the cfDNA concentrations are insufficient for detection. Moreover, differences in the cfRNA expression patterns of the cancerous and healthy organs may reflect functional, longitudinal changes during disease or the treatment of a disease. 

**Table 1 ijms-24-11026-t001:** Strengths and limitations of different liquid biopsy components.

Liquid Biopsy Component	Strengths	Limitation
ctDNA	Analyzing drug effects and predicting acquired resistance [20,21,22,57,73] Minimal residual disease detection [55] The method and data analysis are well established Monitoring the progression of the metastatic process [57]	Low predictive value for mutations occurring in single or small groups [59] Insufficient sensitivity to detect mutations when mutant allele fraction (MAF) is low [58] Lack of standardization of methodology [59] Inability to analyze RNA [59] Difficult to determine the tumor tissue of origin (TOO) [24]
ctRNA	Profiling RNA expression [24] The tissue of origin (TOO) [14,64] Cell-to-cell communication [65,66,69,70,71,72]	Low RNA abundance and instability [67] Difficult RNA extraction [68] Lack of standardization of methodology [68]
CTC	Analyzing molecular and morphological characteristics [42] Profile analysis of DNA, RNA, and proteins [36] Utility as a prognostic or predictive marker [38,39,40,41] Monitoring metastatic process and disease progression [37,38,39,40,42] Functional analysis [36]	Low level of efficiency in blood isolation [43] Instability of CTCs [44] CTC heterogeneity [45] Lack of standardization of methodology [46,47] Inflammatory diseases of the gastrointestinal tract result in false positive results [50] Tumor metastasis and CTC isolation result in false negative results [51,52]
EV	Stable in biological fluids [53] Low immunogenicity [54] Cell-to-cell communication [53]	Variability among isolation techniques; lack of standardization [53] Contamination of non-EV particles and the difficulty of functional characterization [53]

## 3. Application of Circulating RNA Sequencing in Colorectal Cancer

Based on the specific characterization of cfRNAs, it may be possible to develop individualized therapeutic and diagnostic options, which can take into account the interaction between the tumor and TME in the other cancer types. For instance, Raez et al. examined and measured *PD-L1* cfmRNA expression to assess the clinical responses of patients with non-small-cell lung cancer, and they concluded that cfRNA can be used as a tool to predit the onset and progression of cancer [55]. Hieter et al. distinguished cancers from premalignant conditions and predicted disease onset using cfRNA profiles [74]. Likewise, cfRNA, in conjunction with the telomere-specific reverse transcriptase mRNA, was used in another study to measure the responses of cancer patients to chemotherapy, and the results showed that cfRNA could be used to predict and measure the effects of chemotherapy [75,76]. cfRNA has also been shown to contain organ-specific transcripts that can be affected by the development of cancer or tumors [14,77]. Moreover, Li et al. identified potential biomarkers for lung cancer subtypes using circulating cfRNA sequencing. Several differentially expressed genes and pathways associated with lung cancer subtype classification were found to be associated with different lung cancer subtypes. Furthermore, the cfRNA biomarkers were able to predict the outcomes of patients, including their responses to therapy and survival times [78]. The high signal strength of cfRNA is leading scientists to examine whether cfRNA can be used to detect and monitor post-surgical MRD and the disease burden, to predict the recurrence of disease [55] and determine whether it is useful for clinicians to track the treatment response or resistance based on tumor heterogeneity.

CRC biomarker discovery is mostly facilitated by differential gene expression (DEG) analysis of disease groups with healthy controls [79,80]. Samples with different species or stages can be analyzed for circulating RNA sequencing to identify genes that are differentially expressed, revealing their functions and possible molecular mechanisms [14,79]. Recently, circulating RNA sequencing has made a significant contribution to research in various fields, particularly in cancer research, such as cancer diagnostic, prognostic, predictive, subtype classification, and drug resistance applications, involving different novel circulating cfRNA biomarkers in CRC, such as cell-free message RNA (cf-mRNA), cell-free microRNA (cf-miRNA), cell-free long non-coding RNA (cf-lncRNA), and cell-free circular RNA (cf-circRNA) (Figure 1). Supervised machine learning has become increasingly popular in cancer research as a means of identifying and validating diagnostic biomarkers. The idea behind this approach is to use algorithms to learn from labeled data to make predictions or decisions about new, unknown information [81]. The use of supervised machine learning in cancer research involves identifying gene expression signatures from cancer and non-cancer samples, which may then be used to predict the cancer diagnosis or prognosis [81]. On the other hand, unsupervised machine learning is used most often regarding the cancer prognosis and treatment response to identify patterns in large datasets without prior knowledge of the underlying structure, allowing the discovery of novel patterns and associations [81,82]. To identify key characteristics of the data, several dimensionality reduction techniques, such as principal component analysis and independent component analysis, can be used. As another application of unsupervised machine learning in cancer research, the identification of subtypes of cancer based on molecular characteristics can also be used. For example, clustering algorithms can be used to classify patients with similar gene expression profiles, thereby identifying cancer subtypes with distinct molecular characteristics and clinical outcomes.

Cancer biomarker validation using public datasets has become increasingly common, as it provides the opportunity to evaluate the biomarker’s performance in a large number of samples with information on clinical outcomes and patient characteristics. These datasets include the National Center for Biotechnology Information (NCBI), the European Bioinformatics Institute (EBI), and the Cancer Genome Atlas (TCGA). To determine the sensitivity, specificity, and accuracy of the biomarker, a statistical analysis, such as receiver operating characteristic (ROC) analysis, is performed on public datasets. An important step in the validation of biomarkers using public datasets is cross-validation. This involves testing the biomarker’s performance in independent datasets, to ensure that the biomarker does not overfit the original dataset and can be generalized to other datasets by performing this step [83]. Moreover, a comprehensive understanding of the interactions between proteins and the mechanisms underlying various biological processes and diseases can be gained by integrating protein–protein interaction (PPI) networks with the Kyoto Encyclopedia of Genes and Genomes (KEGG).

## 4. Circulating Cell-Free RNA as a Diagnostic Biomarker in CRC

Several studies have demonstrated that various cf-mRNAs are present in the plasma of CRC patients, and their levels are predictive of patient survival [24,31,55,75,79,84,85] (Table 2). Wong et al. were the first to demonstrate that plasma beta-catenin mRNA can serve as a potential marker for CRC [31]. Our team also demonstrated that the downregulation of *RAS* homolog family member A (*RHOA*) and glycogen synthase kinase 3 alpha (*GSK3A*) expression in plasma may function as a diagnostic biomarker of colorectal adenoma using a target sequencing approach [79]. Reduced *GSK3A* expression may be due to a dysfunctional immune response in colorectal adenoma, and inactivation of *RHOA* induces cancer cells to invade and de-differentiate through the Wnt signaling pathway [79]. Additionally, the *SOX9*-based 9-gene panel (*SOX9*, *GSK3A*, *FZD4*, *LEF1*, *DVL1*, *FZD7*, *NFATC1*, *KRT19*, and *RUVBL1*) demonstrated a high level of non-invasive diagnostic performance for CRC (AUC: 0.863) [84]. SOX9 is involved in CRC cell invasion and migration and has been shown to promote tumor growth and metastasis by regulating various signaling pathways, including the Wnt/β-catenin pathway [84]. Prostaglandin-endoperoxide synthase 2 (*PTGS2*), jagged canonical notch ligand 1 (*JAG1*), and guanylate cyclase 2C (*GUCY2C*) mRNA levels in serum and peripheral blood were found to be upregulated in metastatic CRC by ddPCR, while a correlation was found between the serum expression of *GUCY2C* and *GUCY2C*/*PTGS2* and the therapeutic response [86]. Patients with metastatic colorectal cancer (mCRC) had significantly higher levels of *B2M*, *TIMP-1*, and *CLU* mRNAs in their plasma [87]. The combination of the three mRNAs’ levels can be used to discriminate between mCRC and healthy individuals’ plasma, with an AUC of 0.903, 82% sensitivity, and 93% specificity [87]. 

The use of blood serum/plasma miRNAs as diagnostic biomarkers for CRC has been reported by several research groups, as these biomarkers are easy to handle, inexpensive, and can be obtained with minimal invasiveness. Nassar et al. identified miRNA panels (*miR-21*, *miR-145*, *miR-203*, *miR-155*, *miR-210*, *miR-31*, *miR-345*) that may also be useful in diagnosing advanced-stage IV [88]. Fellizar et al. evaluated the expression levels of *miR-21-5p*, *miR-29a-3p*, *miR-92a-3p*, *miR-135b-5p*, *miR-196b-5p*, and *miR-197-3p* in 41 CRC patents and matched adjacent tumor tissue validated with 36 matched plasma samples. *miR-21-5p*, *miR-29a-3p*, *miR-92a-3p*, *miR-196b-5p*, and *miR-197-3p*, but not *miR-135b-5p*, were upregulated in both the CRC tissue and plasma samples, and they can therefore act as potential diagnostic biomarkers of CRC [89]. In Silva et al.’s study, they combined four plasma miRNAs to construct a signature that was able to distinguish between CRC patients and healthy individuals, as well as adenomas and thickened polyps [90]. Recently, the expression of *miR-133a*, *miR-574-3p*, and *miR-27a* was found to differ significantly among different stages, grades, and sizes of CRC, with the combination of these biomarkers showing higher sensitivity for early diagnosis [91].

CRC has been associated with the deregulation of cf-lncRNA transcripts, which impacts primary cancer hallmarks, such as proliferation, apoptosis, invasion, metastasis, and angiogenesis, which have been linked to drug resistance and metabolic disorders [92]. Due to their ability to target multiple pathways disrupted in patients with CRC, lncRNAs are attractive therapeutic candidates. Thus far, there are over 210,000 different lncRNAs that have been identified, and they have been demonstrated to have an important role in the regulation of transcriptional control, splicing, and post-transcriptional progression [93]. Ju et al. demonstrated that the overexpression of lncRNA differentiation antagonizes non-protein-coding RNA (*DANCR*) in CRC, and it also tended to have the worst prognosis and served as a potential diagnostic biomarker for CRC [94]. Akbari et al. showed that the expression levels of *ATB* and *CCAT1* in plasma were significantly upregulated in CRC compared with healthy controls, with sensitivity and specificity of 82% and 75%, respectively [95]. Besides these studies, Radanova et al. demonstrated a significant increase in cf-circRNA expression in the plasma of patients with advanced disease, compared to healthy controls, for the four cf-circRNAs *hsa_circ_0001445*, *hsa_circ_0003028*, *hsa_circ_0007915*, and *hsa_circ_0008717*. Moreover, a panel of *hsa_circ_0001445* and *hsa_circ_0007915* was found to differentiate between patients in stage III and stage IV with high sensitivity and specificity (90.98% and 60.71%, respectively) [96]. 

To further categorize these biomarkers based on their functional mechanisms, mRNA biomarkers such as *SOX9*, *GSK3A*, *GUCY2C*, *B2M*, *TIM-1*, and *CLU* are involved in various pathways that regulate the development and progression of CRC, including the Wnt/β-catenin pathway [84,97], PI3K/AKT/mTOR pathway [84], cyclic guanosine monophosphate (cGMP) signaling pathway [86], major histocompatibility complex (MHC) class I pathway [87], immune response regulation, and apoptotic pathway [87]. *RHOA* and *PTGS2* are involved in the actin cytoskeleton dynamics pathway [79] and the production of prostaglandins [86], respectively. *JAG1* is involved in the regulation of the Notch signaling pathway, which plays a complex role in CRC [98]. 

For the non-coding RNA biomarkers, we can further categorize them based on the tumor development. *Let-7e-5p*, a microRNA, has been demonstrated to function as a tumor suppressor by inhibiting cell proliferation in human cells [99]. It has also been found that *miR-106a-5p*, another microRNA, is dysregulated in patients with CRC and may serve as a useful biomarker for the early detection of the disease [100]. In addition, *miR-133a* has been implicated in the progression of CRC and has been shown to inhibit cell proliferation and cell cycle progression [101]. *miRNA-21*, *miR-29a-3p*, *miR-92a-3p*, and *miR-210* have also been identified as potential biomarkers in CRC, with the dysregulation of these microRNAs associated with tumor progression and metastasis [102,103,104]. The *CCAT1* gene has also been identified as a potential biomarker in CRC, being detectable in all stages of tumorigenesis, as well as the peripheral blood of patients with CRC [105].

**Table 2 ijms-24-11026-t002:** Different diagnostic biomarkers on CRC.

Year	Biomarker	Biomarker Type	Samples	Up-/Downregulated in CRC Patients	Value	Technology	Reference
2019	*RHOA, GSK3A*	mRNA	40 plasma adenoma CRC 39 plasma normal	Down	Diagnosis	Target-Seq	[79]
2023	*SOX9*	mRNA	34 plasma adenoma CRC 19 plasma normal	Up	Diagnosis	Target-Seq	[84]
2021	*PTGS2, GUCY2C, and JAG1*	mRNA	59 serum mCRC	Up	Diagnosis	ddPCR	[86]
2023	*B2M, TIM-1, and CLU*	mRNA	107 plasma mCRC 53 plasma control	Up	Diagnosis	RT-qPCR	[87]
2022	*miR-21, miR-145, miR-203, miR-155, miR-210, miR-31, miR-345*	miRNA		Up	Diagnosis	RT-qPCR	[88]
2022	*miR-21-5p, miR-29a-3p, miR-92a-3p, miR-196b-5p, miR-135b-5p*	miRNA	41 CRC tissue and adjacent tumor tissue 36 plasma CRC	Up Down	Diagnosis	RT-qPCR	[89]
2023	*miRNA-133a, miRNA-574-3p, miRNA-27a*	miRNA	100 serum CRC 20 control	Down Up	Diagnosis	RT-qPCR	[91]
2021	*miR-28-3p, let-7e-5p, miR-106a-5p, and miR-542-5p*	miRNA	109 plasma	Up	Diagnostic	RT-qPCR	[90]
2019	*ATB and CCAT1*	IncRNA	74 plasma CRC 74 control	Up	Diagnosis	RT-qPCR	[95,106]
2020	*DANCR*	IncRNA	40 serum CRC 40 control	Up	Diagnosis	RT-qPCR	[94]
2021	*hsa_circ_0001445, hsa_circ_0003028, hsa_circ_0007915, and hsa_circ_0008717*	circRNA	150 plasma CRC	Up	Diagnosis	RT-qPCR	[96]

## 5. Circulating Cell-Free RNA as a Prognostic Biomarker in CRC

The first study to establish an association between circulating miRNA and prognosis in CRC was published in 2010 [107]. Several promising cfRNAs are associated with overall survival (OS) and poor progression-free survival (PFS) among CRC patients. It is also possible to predict the prognosis of CRC by analyzing the cfRNAs that circulate in the bloodstream and provide insights into the biological behavior and clinical evolution (Table 3). In our recent study, we also found that the combination of 15-hydroxyprostaglandin dehydrogenase (*HPGD*), phosphofurin acidic cluster sorting protein 1 (*PACS1*), and tyrosyl-DNA phosphodiesterase 2 (*TDP2*) expression was associated with survival probability (AUC of 0.838) when using whole-transcriptome sequencing, which supports the idea that these genes may be used as potential prognostic biomarkers for CRC [24]. Pun et al. analyzed postoperative plasma *Bmi1* mRNA levels, which were reduced significantly in patients who did not show a reduction in postoperative *Bmi1* mRNA levels; thus, it can be used to predict distant metastasis and monitor occult metastasis [108]. Moreover, He et al. demonstrated that the presence of high levels of *SSR4* expression in tumor-infiltrating lymphocytes (TILs) may be used as a prognostic biomarker to predict better OS and treatment outcomes in CRC patients; this was demonstrated using bioinformatic analysis on the colon adenocarcinoma (COAD) dataset [109]. Cui et al. demonstrated that the *CXCL3* levels were upregulated in CRC patients, and it is associated with cancer progression and poor prognosis for CRC patients [110]. The presence of high *miR-1290* expression has been associated with an advanced stage and poor prognosis of CRC [106]. According to Kudelova et al., the significant downregulation of *miR-16-5p* was observed and the upregulation of *miR-155-5p*, *miR-21-5p*, and *miR-191-5p* was observed. The same pathways may also contribute to intestinal epithelial regeneration and control wound healing. It has been suggested that the levels of miRNA expression associated with intestinal wound healing and the recurrence of disease are influenced by the levels of miRNA in patients’ circulation [111]. Besides the above, Hao et al. found that stratified stage I to III patients whose plasma *miR-21* level was high had a significantly worse survival outcome when predicting CRC recurrence, indicating that stratified patients have different values when predicting CRC recurrence [112].

## 6. Circulating Cell-Free RNA as a Response to Therapy and Drug Resistance in CRC

Precision therapies have revolutionized the oncology field by targeting specific gene alterations characteristic of specific neoplasms, resulting in the development of new treatment options, such as anti-*EGFR*, anti-*HER2*, and anti-*VEGF* antibodies; immune checkpoint inhibitors; and small-molecule tyrosine kinase inhibitors [113]. However, over time, tumor-induced resistance mechanisms have become more prevalent, making these drugs increasingly ineffective. Therefore, it has become increasingly necessary to identify biomarkers that can provide insights into the emergence of resistance mechanisms. Drug resistance to targeted therapies is influenced by a variety of biological determinants, including the existence of undetectable genomic drivers, mutations in drug targets, the activation of survival signaling pathways, and the inactivation of downstream death signaling pathways [113]. Several studies have demonstrated the potential predictive value of miRNAs as biomarkers for anti-targeted therapy responses in tissue samples [113]. Furthermore, the tumor microenvironment may contribute to resistance via several mechanisms, including promoting the immune evasion of cancer cells, obstructing the absorption of drugs, and stimulating the growth factors of cancer cells [113]. 

cf-miRNAs were first investigated in 2013 as non-invasive biomarkers for the prediction of chemotherapy resistance in CRC [114]. Many studies have been conducted in the last few years to investigate the role of serum and plasma miRNAs in predicting the sensitivity of CRC to chemotherapy [114,115,116]. Patients with chemoresistant CRC were found to have lower levels of *miR-1914-3p* and *miR-1915-3p* in plasma than those with responsive disease. It has been demonstrated that these two miRNAs contribute to the resistance of cells to oxaliplatin and 5-fluorouracil (5-Fu) [116]. Ge et al. also demonstrated that the inhibition of miR-96 increased oxaliplatin sensitivity in CRC cells [117]. Zhang et al. conducted a large-scale validation phase, which resulted in the identification of five miRNAs (*miR-20a*, *miR-130*, *miR-145*, *miR-216*, and *miR-372*) that were significantly downregulated in serum upon exposure to oxaliplatin, which enabled differentiation between primary sensitive and resistant patients, demonstrating the value of this panel in selecting a treatment for CRC [118]. In another study, Ye et al. demonstrated that the lncRNA *GMDS-AS1*’s direct target, HuR, is constitutively activated by STAT3/Wnt signaling and plays an important role in the development of CRC tumors; thus, it can be used to diagnose, monitor, and predict CRC outcomes [119]. The downregulation of plasma lethal-7 (*Let-7*) miRNA levels has been observed in CRC patients, and there was a significant association between higher plasma levels of *let-7* and CRC patients. This can improve survival outcomes regardless of the mutational status of *KRAS* and can provide insights into the patient population that responds to anti-EGFR therapy [120,121]. There is an association between the expression of *miR-31-5p* and shortened PFS in patients with metastatic CRC treated with anti-EGFR therapy [122]; thus, it has demonstrated a strong diagnostic ability for CRC in serum [123]. Schou et al. also demonstrated that cetuximab and irinotecan failed to elicit a response in patients with high plasma *miR-345* expression [124]. 

## 7. Challenges in Circulating RNA Sequencing in Plasma

Circulating RNA sequencing in plasma has emerged as a promising method for the identification of disease biomarkers and potential therapeutic targets. According to the literature review, there are several challenges associated with the use of cfRNA analysis, which can impact the accuracy and reproducibility of the results. 

The technical challenges, low cfRNA quantities, RNA degradation, heterogeneity of plasma samples, RNA fragmentation, contamination, lack of standardization, and bioinformatics challenges are the main challenges. Pre-analytical parameter setting, as well as robust and standardized protocols to optimize the potential of RNA-seq for plasma cfRNA analysis, is important to overcome these challenges. Several pre-analytical parameters can influence the ctRNA detection sensitivity, which is strongly dependent on the quantity and quality of the input material. Precaution is essential to prevent cfRNA degradation and contamination by fat, genomic DNA, Ca2+, proteins, cell debris, and exogenous contaminants, such as glove powder [125]. gDNA contamination can affect the concentration level of RNA in plasma/serum, which could affect the accuracy of the final results because of the large number of cellular proteins, as well as cellular debris [126]. Other pre-analytical factors also influence the limit of detection and accuracy of the analysis, including the nature of the anticoagulant in the blood collection tube, plasma volume, preservation approaches, and extraction approaches. The use of different blood collection tubes has been evaluated in several studies [127,128,129]. A pre-coated sample tube with preservatives is used to avoid cell lysis, thereby preventing the release of cell-free nucleic acid from hematopoietic cells. EDTA tubes should not be centrifuged for more than four hours after sampling [128]. Alternatively, other blood collection tubes containing stabilizing agents, such as cell-free RNA Streck tubes, can be stored at room temperature for several days without adversely affecting the results of subsequent analyses [129]. 

Following the collection of blood, the tubes must be centrifuged to separate the plasma. The centrifugal force was shown to cause the inconsistent quantification of mRNA in plasma by Wong et al. [130]. Numerous studies have attempted to determine the best centrifugation protocol to determine the optimal concentration of cfRNA during this step [80,130,131]. Following our first study examining the effect of the centrifugal force on plasma cfRNA by RT-qPCR, we validated two-step centrifugation protocols using target mRNA sequencing, which was found to be the most suitable for the prevention of undesirable genomic DNA, reducing the degradation of cfRNA and efficiently removing RNA-associated particles, such as cell debris, from plasma [131]. To avoid cell lysis, blood cells must be centrifuged slowly (1600× *g* for 10 min at +4 °C) first. Following this, cellular debris and fragments can be removed from the plasma supernatant by short-term high-speed centrifugation (16,000× *g* for 10 min at +4 °C). While collecting the plasma after the first spin, it is critical not to disrupt the buffy coat. The plasma samples should be preserved with Trizol LS reagent before long periods of storage at −80 °C, which is used for the isolation of total RNA from cells by disrupting and dissolving cell components while maintaining the integrity of the RNA at the same time [132].

An individual transcriptome snapshot can be approximated with cfRNA obtained from blood samples. However, diverse methodologies for cfRNA isolation are subject to biases that may obscure any useful biological information. Currently, commercial column-based kits are more widely used than traditional methods, such as phenol–chloroform and guanidium–thiocyanate. Wong et al. optimized the extraction of non-viral mRNA from plasma using Trizol LS and the RNeasy kit together, resulting in the isolation of all RNA molecules larger than 200 nucleotides [132]. The traditional approach favors the isolation of selective RNA populations, which often leads to a decrease in the amount of cfRNA produced [133]. It has been reported by Kim et al. that biological samples with low levels of RNA cannot be extracted with the guanidium–thiocyanate approach, due to GC-poor or highly structured miRNAs [134]. However, the technical differences related to the kit-dependent biases will provide varying levels of plasma cfRNA content [135]. Therefore, it is important to select the best method for the isolation of RNA based on the study’s end goal. Meanwhile, DNA contamination is a major concern in cfRNA isolation, and an additional step to treat the sample with DNAse before [79,131] or after extraction [136] is necessary. Alternatively, several studies show that using carriers, such as glycogen, can increase the RNA yield [137]. To assess the efficiency and reproducibility of an extraction method, so-called spike-in controls can be used to spike the starting material with artificial or exogenous ribonucleotides and quantify their recovery [137]. For the extraction of samples with low RNA content and small RNA species, it is also suggested to add MgCl2, to stabilize the RNA–RNA interaction [137].

A major challenge in the bioinformatics analysis of circulating RNA sequencing data is the quality of the sequencing reads. The reads obtained from sequencing data are often short and of low quality, which can result in errors in the alignment and quantification of the reads [138]. Moreover, circulating RNA sequencing data may contain contaminants and biases due to the presence of extracellular vesicles and other cellular debris in plasma samples [139]. Identifying circulating RNA biomarkers is another challenge. It is necessary to integrate multiple datasets from different studies and to use complex statistical models to identify circulating RNA biomarkers [140]. Additionally, the lack of standardized pipelines for the analysis of circulating RNA sequencing data is a significant challenge. In the absence of standardization, different bioinformatics pipelines may yield different results [141]. Finally, interpreting data from circulating RNA sequencing can be challenging. To verify the biological significance of the identified circulating RNA biomarkers, functional studies need to be performed, which are time-consuming and expensive.

## 8. Future Perspectives

Future research perspectives on tumor heterogeneity in CRC patients and cfRNA research will require further advancements in technology, as well as the integration of multi-omics approaches. A combination of single-cell sequencing (scRNA seq) and transcriptome sequencing can provide a more detailed understanding of the genetic, genomic, and transcriptional heterogeneity within tumors. For instance, Jin et al. used published single-cell transcriptomics datasets to deconvolute the cell type abundance among paired plasma samples from CRC patients who underwent tumor-ablative surgery, to identify tissue-specific contributions to the cfRNA transcriptomic profile, and they found that intestinal secretory cells were downregulated after surgery [24]. Besides this, an analysis of transcriptomic profiles by scRNA seq provides a comprehensive view of the cellular activity within the TME, as well as their interactions with each other. Through scRNA seq, the molecular and genomic profiling of high-quality and high-quantity immune cells can be performed and cellular heterogeneity in the cancer microenvironment can be assessed [142]. Advances in technology, such as next-generation sequencing (NGS) and single-cell transcriptomics, enable a more comprehensive analysis of intra-tumor heterogeneity. The focus of future research should be on integrating multi-omics approaches and exploring the dynamics of tumor heterogeneity in order to identify novel therapeutic targets and to improve personalized treatment strategies for CRC.

Many researchers have recently discovered that analyzing the patterns of DNA fragment sizes in the blood allows them to identify cancer as well as its location in the body [143]. A better understanding of cfDNA fragmentation has provided several fragmentomic markers, including fragment sizes, preferred ends, end motifs, single-stranded jagged ends, and nucleosomal footprints [144]. The application of “fragmentomics” to cfRNA remains controversial, given that cfRNA is highly degraded and comparatively unstable compared to cfDNA and may be cleaved by members of the ribonuclease A (RNases) superfamily, resulting in different length distributions. Therefore, a comprehensive understanding of cfRNA fragmentation patterns holds significant potential in advancing our understanding of a variety of biological processes as well as diseases. Identifying the fragmentation patterns of cfRNA can provide valuable insights into the mechanisms by which RNA is degraded, the stability of RNA, and the functional consequences of RNA fragmentation. It is essential to leverage advanced technologies and analytical approaches to fully explore the future perspectives of cfRNA fragmentation patterns. With high-throughput sequencing techniques, such as RNA-seq, cfRNA fragmentation patterns can be comprehensively and unbiasedly profiled [145]. Moreover, the integration of bioinformatics tools and machine learning algorithms can facilitate the identification of specific fragmentation patterns and their functional significance.

The clinical implementation of cfRNA-based assays relies on the crucial identification of specific cfRNA molecules or fragmentation patterns consistently associated with cancer diagnosis, prognosis, and treatment responses.

Future perspectives in this field hold great promise in improving cancer diagnostics, monitoring treatment responses, and advancing personalized medicine. The standardization of liquid biopsy workflows and the optimization of pre-analytical conditions, including standardized protocols for blood collection, storage, and processing, are essential to ensure the reproducibility and reliability of cfRNA analysis. Additionally, the development of robust and sensitive techniques for cfRNA extraction, library preparation, and sequencing is crucial to obtain accurate and consistent results. The integration of cfRNA analysis with other liquid biopsy components, such as ctDNA and CTCs, can provide a more comprehensive and informative picture of the tumor’s molecular profile, enhancing the sensitivity and specificity of liquid biopsy assays and improving their clinical utility in cancer management.

## 9. Conclusions

In conclusion, the role of cfRNA in the development of CRC has been extensively studied. CRC biomarkers, including both coding and non-coding RNAs, have emerged as promising tools for diagnosis, prognosis, prediction, and monitoring. By comparing the strengths and limitations of different liquid biopsy components, we can improve our understanding and make corresponding decisions or accurately select strategies, in order to improve their clinical utility in cancer management. The heterogeneity of patient populations, small sample sizes, the availability of samples, and the time-consuming and expensive nature of the methods are the most significant barriers that hinder researchers’ decisions to pursue prospective validation. It is therefore essential to identify unique circulating signatures in CRC that are specific and sensitive by utilizing a standardized, consistent approach throughout the entire research process, including blood collection, plasma preparation, handling, storage, and extraction and quantification, to minimize the risk of conducting a costly and time-consuming prospective validation study that may result in unusable results. The characterization of the TME and the understanding of its function and formation will contribute to an improved understanding of its potential as a CRC biomarker in the future.

## Figures and Tables

**Figure 1 ijms-24-11026-f001:**
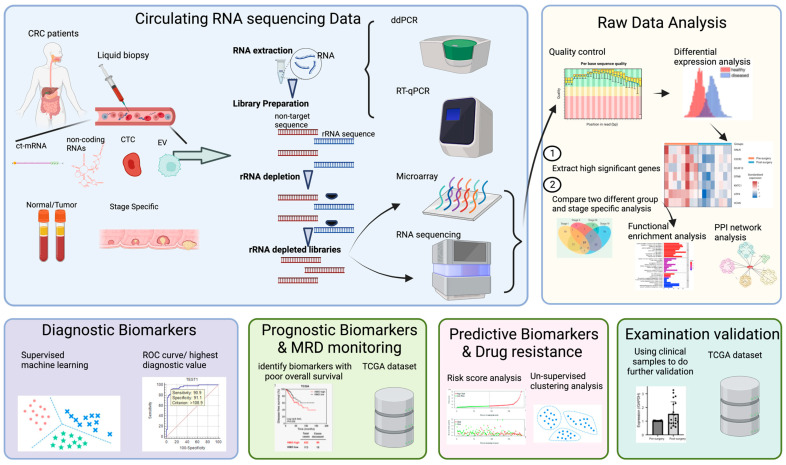
Workflow and application of circulating RNA sequencing (created with BioRender.com, accessed on 20 May 2023).

**Table 3 ijms-24-11026-t003:** Different prognostic biomarkers for CRC.

Year	Biomarker	Biomarker Type	Samples	Up-/Downregulated in CRC Patients	Value	Technology	Reference
2023	*HPGD, TDP2, PACS1*	mRNA	8 plasma pre-surgery CRC and post-surgery CRC; 8 tumor tissue and adjacent tumor tissue	Down Up	Prognosis	Transcriptome Seq	[24]
2021	*SSR4*	mRNA		Up	Prognosis	Data Mining	[109]
2022	*CXCL3*	mRNA	228 CRC vs. 216 control	Up	Diagnosis and Prognosis	RT-qPCR	[110]
2014	*Bm1*	mRNA	45 CRC	Down	Prognosis	RT-qPCR	[108]
2022	*miR-1290*	miRNA		Up	Prognosis	ddPCR	[106]
2022	*miR-155-5p, miR-21-5p, miR-191-5p* *miR-16-5p*	miRNA	110 plasma CRC	Up Down	Prognosis	RT-qPCR	[111]
2022	*miR-21*	miRNA	113 CRC	Up	Prognosis	RT-qPCR	[112]

## Data Availability

No new data were created or analyzed in this study. Data sharing is not applicable to this article.

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
