# Peer review of "Exploring the Role of Circulating Cell-Free RNA in the Development of Colorectal Cancer"

_ijms, 2023, doi:10.3390/ijms241311026_

Round 1

Reviewer 1 Report

The manuscript by Kan et al. reviewed the role of circulating cell-free RNA (cfRNA) as biomarkers for the diagnosis, prognosis, prediction, and treatment of colorectal cancer (CRC). Overall, I found this manuscript is well organized and provides a thorough review of circulating cfRNA in CRC.

One major concern is that a lot of content in this manuscript is just a list of different studies. It would be great if the authors could put more effort on the biological questions.

Minor concerns:

1.       The title includes “the development of colorectal cancer”. Please provide some background on the development CRC

2.       The authors provide some examples on circulating cfRNA as the biomarkers for diagnosis, prognosis of CRC. I believe it would help a lot if the authors summarize these cfRNA into different categories based on their function mechanisms.

The English is ok.

Reviewer 2 Report

The review covers an important and timely topic. The manuscript reads well. However, the manuscript suffers from two major points 1) a proper citation at different places, and 2) the Conclusion & Future Perspectives section is not presenting consistent with the review aim and does not show aclear future perspectives    Examples for inappropriate citations   lines 143-155, this paragraph needs a careful citation. This paragraph proposes that RNAs establish a link between cancer cells and TME without giving a single relative citation to confirm this hypothesis.
in lines 144-147, the cited papers are not related to this text.   Table 1. The citations should be carefully placed for the advantages and limitations of each liquid biopsy component. Referring only to review articles is not a proper way.   For the Conclusion & Future Perspectives section, I do not understand what the author would like to deliver. The prosectives are vaguely defined. There are missing connections between sentences. They start with the role of tumor heterogeneity inTME, passing with precision medicine, to transcriptome analyses without a clear connection. The review scope is not about tumor heterogeneity, the Future Perspectives should be consistent with the aim of the review.  I suggested rewriting this important section. 

Although the manuscript reads well, but at a few places there are some sentences are not understandable. I suggest proofreading for the manuscript

Round 2

Reviewer 1 Report

The authors addressed my concerns. I don't have further concerns.

English is ok